# The impact of differences in text segmentation on the automated quantitative evaluation of song-lyrics

Friederike Tegge[1]*, Katharina Parry[2]

1 School of Humanities, Massey University, Palmerston North, New Zealand, 2 School of Fundamental Sciences, Massey University, Palmerston North, New Zealand

☯ These authors contributed equally to this work.

* friederike.tegge@gmail.com

**Data Availability Statement:** All relevant data are within the manuscript and its Supporting information files.

**Funding:** FT $1000 Massey University www. massey.ac.nz. The funders had no role in study

## Abstract

The text-evaluation application Coh-Metrix and natural language processing rely on the sentence for text segmentation and analysis and frequently detect sentence limits by means of punctuation. Problems arise when target texts such as pop song lyrics do not follow formal standards of written text composition and lack punctuation in the original. In such cases it is common for human transcribers to prepare texts for analysis, often following unspecified or at least unreported rules of text normalization and relying potentially on an assumed shared understanding of the sentence as a text-structural unit. This study investigated whether the use of different transcribers to insert typographical symbols into song lyrics during the pre-processing of textual data can result in significant differences in sentence delineation. Results indicate that different transcribers (following commonly agreed-upon rules of punctuation based on their extensive experience with language and writing as language professionals) can produce differences in sentence segmentation. This has implications for the analysis results for at least some Coh-Metrix measures and highlights the problem of transcription, with potential consequences for quantification at and above sentence level. It is argued that when analyzing non-traditional written texts or transcripts of spoken language it is not possible to assume uniform text interpretation and segmentation during pre-processing. It is advisable to provide clear rules for text normalization at the pre-processing stage, and to make these explicit in documentation and publication.

## Introduction

Automated evaluation of language and text is becoming increasingly sophisticated, with a range of programmes now available to examine text at the word level, sentence level, and even discourse level. Coh-Metrix is an application "at the forefront of these technologies" [1], offering a broad range of measures of local and global, lexical and textual discourse features. Its usefulness is demonstrated by its popularity with applications ranging from research on schizophrenia [2] to higher education [3].

While Coh-Metrix has most often been used for the analysis of printed text, it can, according to its creators, also be used to analyse spoken discourse and non-traditional written texts

design, data collection and analysis, decision to publish, or preparation of the manuscript.

**Competing interests:** The authors have declared that no competing interests exist.

with "untidy language and discourse" [4] such as typed chat exchanges, instant messaging, emails, and poetry. While certain features frequently found in spoken and non-traditional written texts, including disfluencies and ungrammatical utterances, might impact some of the indices available in Coh-Metrix, the authors assert that "many of the measures are minimally disturbed".

The present study is part of a wider investigation of English pop song lyrics and their suitability for teaching and learning English as a second language (ESL). Song lyrics usually deviate from formal formatting and editing standards of written texts, for example in their text layout, punctuation, and spelling, and they often contain song-specific disfluencies and interjections. In particular, explicit indications of sentence boundaries in lyrics are often omitted. A further complication lies in the presentation of lyrics in verse format where line breaks frequently occur in the middle of sentences and typically do not indicate the end of sentences or paragraphs but rather signal musical groups and melodic structure (although these can correspond to linguistic constituents).

Given these deviations from widely accepted standards of text formatting, song lyrics might require principled text normalisation, that is, the conversion "to a more convenient standard form" [5] prior to their automated textual assessment. The goal of this study is to understand the potential impact of text normalisation at the pre-processing stage, specifically when it comes to the identification of sentence boundaries, on the results of a Coh-Metrix analysis of English pop song lyrics.

## 1 The relevance of the sentence in Coh-Metrix 3.0

Natural language processing (NLP) and automated text evaluation frequently rely on the sentence as an underlying unit, and text normalisation often "includes sentence segmentation: breaking up a text into individual sentences using cues like periods or exclamation points" [5]. It appears that Coh-Metrix 3.0—the most recent version available—also relies on the sentence as a basic unit of text segmentation for a large number of its measures. Several descriptive indices specifically focus on sentences in the target texts or use sentences to measure larger units of text, including sentence count (DESSC), sentence length/mean number of words (DESSL), sentence length standard deviation (DESSLd), paragraph length/mean number of sentences (DESPL), and paragraph length standard deviation (DESPLd). Examples are listed in [4] and on www.cohmetrix.com.

Coh-Metrix also assesses levels of lexical information of target texts, their syntactic complexity and density, and dimensions of text cohesion. Text cohesion in this context refers to text-inherent linguistic, semantic, and discourse features that affect the "connectedness of concepts presented in a text" [4] and have an effect on text difficulty and reading (or listening) comprehension. It can be speculated that lexical information such as diversity measures, word frequency, word meaningfulness and concreteness, would be largely unaffected by sentence segmentation, whereas certain syntactic indices such as sentence syntax similarity of adjacent sentences (SYNSTRUTa) clearly rely on the sentence as a unit. Similarly, cohesion measures such as noun-, argument-, stem-, and content-word overlap in adjacent sentences or across all sentences in a text require sentence boundaries to be indicated, as do measures of Latent Semantic Analysis (LSA), that is, measures of relative semantic overlap between sentences and paragraphs. The same holds for traditional readability scores included in Coh-Metrix 3.0, the Flesch Reading Ease (RDFRE) and the Flesch-Kincaid Grade Level (READFKGL), which both rely on sentence length (mean number of words per sentence) to compute text difficulty. The Coh-Metrix L2 Readability score (L2), intended to assess the suitability of texts for second

language learners, includes content-word overlap in and syntactic similarity of sentences as variables in its formula.

Finally, Coh-Metrix 3.0 offers so-called Text Easability Principal Component Scores, based on a principal components analysis (PCA) of 54 indices in a corpus of written texts found across school-grade levels (K-12) and academic subjects (see [4, 6] for further details). Five of these principal components—narrativity, syntactic simplicity, word concreteness, referential cohesion, and deep cohesion, account for 54% of the variance in the test corpus and are provided to Coh-Metrix users as comprehensive scores that are more easily accessible and of more immediate practical use for teaching practitioners. [6] show that sentence-based measures contribute to three of these easability scores (narrativity, syntactic simplicity, referential cohesion).

## 2 The role of punctuation in text segmentation

It appears that in Coh-Metrix 3.0 (and NLP more widely) sentences are not detected based on semantic or syntactic features but rather by means of punctuation. In other words, the end of a sentence is indicated by a full stop, question mark, or exclamation mark; and the sentence unit is detected through the presence of these typographical symbols, which are inserted by the text authors themselves or by members of the research team during the pre-processing phase, based more or less on commonly agreed-upon rules of their use.

One issue, however, is that there is no simple definition of what a sentence is [7, 8]. A sentence is generally understood in structural terms as the largest complete grammatical unit which can be analysed into structural constituents or phrases and which is not connected to other units by means of grammatical relationships [9–11]. However, various linguistic phenomena such as sentence fragments challenge this definition. Particularly in spoken language, sentences are difficult to identify. The more common unit applied to oral discourse is that of utterance, a "behavioural unit" that can refer to single words as well as to longer stretches of speech and which is characterised by features such as prosody and intonation, which may or may not correspond to sentence segmentation and punctuation in written language. As [11] emphasise: "The sentence is an indeterminate unit in the sense that it is often difficult to decide, particularly in spoken language, where one sentence ends and another begins". [12] use the term sentence only for written language, highlighting the reliance on punctuation when arguing that a sentence "is the highest unit of punctuation on the graphological rank scale and has evolved in the writing system (. . .) We will use the term sentence to refer only to this highest-ranking graphological unit of punctuation" (p. 436).

NLP generally, and text assessment involving Coh-Metrix specifically encounter three potential problems: First, sentences are typically identified based on conventions for written texts and their typographical manifestation, that is, punctuation. As [5] explain, "the most useful cues for segmenting a text into sentences are punctuation, like periods, question marks, exclamation points" (p. 29). Second, an utterance in spoken language is frequently incorrectly understood as "the spoken correlate of a sentence" (p. 19). And third, text normalisation in NLP seems to frequently involve inserting punctuation—if missing—without clear documentation of principles followed during the process. This might not be a problem when the target data consists of written texts following formal standards of text composition. However, we hypothesise that the same process applied to non-standard written texts, including song lyrics, characterised by a range of disfluencies and non-standard text formatting might result in a significant but often unacknowledged influence of the transcriber or "punctuator" on the data. [12] touch on the frequent lack of clear guidelines when observing that "many writers punctuate phonologically rather than grammatically, or in some mixture of the two" (p. 8).

Consequently, depending on the genre, style, and mode of production of the target texts, inserting punctuation might require a clear, principled approach followed by all involved in text normalisation in the pre-processing stage.

## 3 How do other studies approach this problem?

How is the problem of punctuation dealt with in other studies that have used Coh-Metrix for data analysis? This question is of particular relevance for research investigating texts that lack sentential segmentation in the original and thus require punctuation by the researchers, including spoken discourse and non-standard written texts. [13] used Coh-Metrix to explore differences in the lyrics written by suicidal and by non-suicidal songwriters. They investigated 560 songs written by 16 songwriters (35 songs each), focusing on 14 indices available through Coh-Metrix 2.0 and the LIWC program (Linguistic Inquiry and Word Count), including LSA and argument overlap in adjacent sentences. Lyrics were "obtained from websites devoted to that artist or band" (p. 1219). It remains unclear whether the websites were curated by the music producers or by fans who do not necessarily have access to the original text. Furthermore, the authors do not specify whether the lyrics presented on these websites included punctuation. They do, however, concede that the websites were not entirely reliable in that they did not necessarily include all repetitions of choruses and phrases as they occurred in the audioversions; accordingly, the lyrics were checked against recordings. Further relevant information is left unspecified regarding who and how many transcribers undertook the text normalization and what guidelines they followed, including how they dealt with fade-outs and non-semantic vocables and how sentences and paragraphs were delineated.

[14] used Coh-Metrix to analyse how linguistic features differ in written and in spoken online communication in multi-party collaborative learning environments. They investigated the impact of modality (written chat, spoken chat) on five principal components (narrativity, referential cohesion, situation model cohesion, syntactic simplicity) and three superficial text characteristics (total number of words, total number of sentences, and number of words per sentence). The description of the transcription process in the extended version of the proceedings is limited to one sentence: "Human transcribers converted conversations in the spoken condition to text for analysis". However, the impact of text normalisation might be of particular interest in this study, as it depends on two different parties for the two modalities in question, that is, on the interlocutors themselves in the written chat mode and on an unspecified number of transcribers in the spoken mode. It also seems to be of relevance how the transcribers were trained, since written chat texts from the participating interlocutors were available and might have served as the basis for the transcribers' decision-making processes while preparing the spoken texts for analysis.

[1] investigated potential linguistic indicators in another non-traditional written genre, truthful and deceptive text messages, using both Coh-Metrix and LIWC with a number of measures directly or indirectly relying on the sentence unit. The corpus consisted of 242 transcripts of instant messaging exchanges, derived from a study conducted by [15]. [15] provide a brief description of the pre-processing of the transcripts: "First, given that CMC [computer-mediated communication] participants often omit punctuation, periods were placed at the end of all turns. Similarly, if a question mark was omitted after a question, a question mark was inserted. (. . .) Last, any misspellings were corrected . . ." (p. 11). The example transcripts in the appendix of [15] indicate that a turn never exceeds the length of one sentence. However, this is conjecture for the transcripts not included in the appendix.

[16] focused on CMC and compared language and cohesion of written "monologues" (textbook and textoids) and written interactional dialogues between students and either human

tutors or an automated tutoring system (see [17]). The interactional dialogues were taken from a previously published experiment ([18]). It can be assumed that the students and human tutors followed written standards of punctuation. Neither [16] nor [18] detail the pre-processing of data; however, one example transcript provided in [18] (p. 16) includes a student response that indicates that data preparation might have been necessary: "That will depend on the time that the ball is in the air/distance ball traveled."

The need for text normalisation, sentence segmentation, and the insertion of punctuation during pre-processing becomes more apparent when research is focused on spoken texts. [19] explored differences between written and spoken registers in partial replication of [20]'s investigation. However, while [20] focused on word-level features, [19] included global text features in their analysis, which rely on sentential segmentation, including coreference, LSA, and readability scores. To compare results, they used the same corpora as [20]; that is, in both studies the spoken section consisted of six registers from the London-Lund Corpus of Spoken English, including among others face-to-face conversations and telephone conversations. While [19] mention that "all textual coding other than alphanumeric characters and punctuation was removed" (p. 845), they do not specify if and how punctuation was inserted into the transcripts of the spoken texts in preparation for the Coh-Metrix analysis. Documentation detailing the design of the London-Lund corpus of spoken English (see [21]) shows that the transcription of spoken texts was based on tonal units and tone unit boundaries and does not originally contain typographic punctuation to delineate sentences.

[22] investigated text-inherent factors influencing human assessment of speaking proficiency in the TOEFL-iBT© (the computer-delivered Test of English as a Foreign Language) and the variance among different scorers. They used Coh-Metrix as well as two further programs, CPIDR (the Computerized Propositional Idea Density Rater; see [23]) and LIWC, to analyse 244 speech samples from a public TOEFL-iBT© data set. Documentation of the transcription process is brief: Samples were transcribed by one trained transcriber and later reviewed by a second transcriber for accuracy. Only lexical words, including disfluencies such as false starts, were included in the transcriptions, whereas metalinguistic data and filler words were not included. [22] state that "periods were added to the samples at the end of idea units" (p. 177). This provides some insight into how the transcriptions were produced. However, several potentially relevant questions remain unanswered. For example, the concept of "idea unit" is undefined, and it is not clear whether the transcribers were trained and followed explicit rules of text segmentation and punctuation. Furthermore, no example transcript is provided. Consequently, an exact replication of the study seems difficult to achieve.

[24] used speech samples from the same TOEFL-iBT© public use dataset to investigate the test-takers' integration of lexical and cohesion features from a listening text into their spoken responses and the effect of such lexical integration on the ratings of speaking proficiency given by human experts. The description of the transcription process was identical to [22]. Finally, [25] also used a data set of TOEFL-iBT© speech samples to investigate an automated speech scoring system and its prediction accuracy of human ratings for speaking proficiency and cohesion in speaking. Again, no example transcript is provided and description of the transcription process is limited to a brief statement: "The spoken responses were all transcribed by humans with punctuation and capitalization" (p.815).

## 4 Methods

As song lyrics are frequently published in a format that requires text normalisation at the pre-processing, we addressed the following questions before proceeding with the wider investigation:

1. Can the use of different transcribers to insert punctuation into pop song lyrics result in significant differences in sentence segmentation?

2. Could this potential variance in sentence segmentation significantly affect Coh-Metrix measures?

To answer these questions, we compiled a small corpus of 30 songs. In an attempt to imitate realistic conditions within a linguistics research group and to reflect the practices described in previous studies involving Coh-Metrix, three different language professionals inserted punctuation into the target text files based on their knowledge of writing conventions and their understanding of the syntax and semantics of the lyrics in question. We then conducted a Coh-Metrix analysis, focusing on any potential differences between transcribers, or rather punctuators, in relevant descriptive measures that underlie many of the more sophisticated measures.

We note that the sample of 30 songs and three annotators is small. The work presented here is a first step in a larger project assessing the readability of pop songs for ESL learners. It is intended to address research-methodological problems that seem to be highly relevant not only to the wider investigation at hand but also to the use of Coh-Metrix and automated text evaluation more widely.

## 4.1 The corpus

30 songs, with a total of 14,287 running words (tokens), were randomly selected from a larger corpus, the Wellington Corpus of Popular Songs ([26]). The WOP comprises 408 English-language pop songs from the top 100 end-of-year US billboard charts from the years 2014, 2012, 2010, and 2008.

The ten pop songs, comprising a total of 4,282 tokens (median: 403), were: Whataya want from me (Adam Lambert; P1), Meet me halfway (The Black Eyed Peas; P2), Boom clap (Charli XCX; P3), Crush (David Archuleta; P4), I'm yours (Jason Mraz; P5), I kissed a girl (Katy Perry; P6), Moves like Jagger (Maroon 5; P7), Cooler than me (Mike Posner; P8), What makes you beautiful (One Direction; P9), Blow me (Pink; P10).

The ten rap songs, comprising 6,682 tokens (median: 662), were: Dance ass (Big Sean; R1), Hot boy (Bobby Shmurda; R2), So good (B.O.B.; R3), Get like me (David Banner; R4), All I do is win (DJ Khaled; R5), Mercy (Kanye West; R6), Mrs Officer (Lil Wayne; R7), Superstar (Lupe Fiasco; R8), Whatever you like (T.I.; R9), My hitta (Yg; R10).

The ten country songs, comprising 3,314 tokens (median: 332), were: Good girl (Carrie Underwood; C1), Springsteen (Eric Church; C2), Wanted (Hunter Hayes; C3), Lover lover (Jerrod Niemann; C4), American kids (Kenny Chesney; C5), Somethin bout a truck (Kip Moore; C6), I don't dance (Lee Brice; C7), Pontoon (Little Big Town; C8), The house that built me (Miranda Lambert; C9), Mine (Taylor Swift; C10).

Note that P, R, and C stand for pop, rap, and country and are used to identify the individual songs in the plot of the correspondence analysis below.

## 4.2 Data collection

The lyrics were obtained from lyrics websites and were checked manually for errors. Lyrics uploaded to these websites are typically provided by music consumers rather than the music producers themselves. Consequently, sentence- and paragraph-segmentation are often omitted or are reliant on the consumer's intuition. In the age of streaming (rather than the publication of recordings accompanied by booklets) lyrics provided by the songwriters themselves are increasingly difficult to find.

If available, they still frequently lack punctuation. The lyrics contained several genre-specific features that needed to be addressed in a principled manner. Firstly, so-called fade-outs, that is, the repetition of (parts of) the chorus or verses at the end of the song at decreasing volume, were removed. Secondly, the corpus contains repetitions of individual words, phrases, choruses, and verses as used in the selected version of the song, including fully audible echoes of phrases and words vocalized by background singers.

An example of word repetition can be found in B.O.B.'s So good: "... pack your bags real good, baby, 'cause you'll be gone for a while, while, while, while." Finally, so-called marginal words (e.g. wobbledy, boing, yo, huh, ha) as well as non-lexical vocables, that is, words that do not have lexical content but are pronounced for the sake of vocalization itself without expressing semantic meaning [27] remained part of the corpus. An example of non-lexical vocables can be found in Lupe Fiasco's Superspar: "And they wanna know, oh oh oh oh, if you are what you say you are, a superstar."

Before the lyrics were given to the punctuators, all punctuation and sentence-initial capitalization was removed by one researcher. This was done so that punctuators would rely on their own knowledge of writing standards rather than on the text segmentation provided by the song consumer who uploaded the text to a lyrics website.

The following excerpt from Eric Church's Springsteen with punctuation/capitalization removed provides a brief example of what the song lyrics provided to the punctuators looked like:

*to this day when I hear that song I see you standin' there on that lawn discount shades store bought tan flip flops and cut-off jeans somewhere between that setting sun I'm on fire and born to run you looked at me and I was done*

What constitutes a paragraph in song lyrics is also contentious. We approached this problem by inserting paragraph breaks based on distinct changes in melody (based on one researcher's intuition during listening). The punctuators were instructed to change paragraphing based on their own (semantic or structural) intuition.

**4.2.1 Instructions for the punctuators.** The following instructions were given to the punctuators:

*Below, you will find the lyrics of 10 songs in verse form. Please indicate for all song lyrics where you would put punctuation, in particular, full stops (or equivalent "sentence-final" symbols like ? and !).*

*For example, is it: "Hey, good girl with your head in the clouds, I bet you I can tell you what you're thinking about." Or is it: Hey, good girl with your head in the clouds! I bet you I can tell you what you're thinking about." Or ...*

*You can use full stops, question marks, exclamation marks, and colons. You can also use commas—but don't spend too much time on those. The paragraphs indicate breaks or melody changes in the singing but they do not necessarily indicate the end of sentences. If you think that a paragraph break actually broke up a continuing sentence, please indicate that as well. Simply delete the paragraph break and "connect" the sentence or sentences. Please do this for the entirety of the lyrics provided here. Please go with your gut feeling. There is no right or wrong answer. This is NOT testing you. Symbols you may use: . ? ! : ,*

**4.2.2 About the punctuators.** Three research assistants were tasked with sentence segmentation and paragraphing of the target lyrics. All three punctuators (here named DET,

LEM, WAT) are native speakers of New Zealand English and language professionals with many years of work experience, albeit in different capacities.

LEM has worked as a journalist and proof-reader for 30 years, holds a BA in linguistics and is currently pursuing an MA in applied linguistics.

WAT has worked in the adult education sector for over eight years with a focus on literacy and numeracy. WAT holds a certificate in adult education (literacy and numeracy) and is currently pursuing a BA in linguistics and an additional subject in the humanities.

DET has five years of work experience as a language teacher (ESL, English for Academic Purposes, Spanish) and two years of experience as a tertiary tutor and marker in language, linguistics, and communication courses. DET holds an MSc in applied linguistics and second language acquisition and is currently pursuing a PhD in applied linguistics.

This project has been reviewed and approved by the Massey University Human Ethics Committee: Southern B, Application 19/49.

## 5 Results

At first glance, there were noticeable differences in the sentence segmentation between the punctuators. An illustration of these differences is provided in Table 1.

The question is, how different is different? We address this question with some statistical analyses with a focus on the number of sentences as a descriptive measure underlying many other, more sophisticated Coh-Metrix measures.

All statistical analyses were performed using the statistical programme R ([28]).

### 5.1 Poisson generalized linear model

As the number of sentences, our characteristic of interest here, are counts, we assume they are Poisson distributed. Thus, we fit a Poisson generalized linear model with the total number of sentences (denoted as NoSen) used to examine the two predictor variables, punctuator and song genre. The two predictor variables are categorical, with three levels each. That is, Punctuator has levels DET, LEM and WAT and Genre has levels COU, POP and RAP for country, pop, and rap:

$$log(NoSen) = \beta_0 + \beta_1 Punctuator + \beta_2 Genre.$$

This model allows us to evaluate the level of difference in sentence counts due to pre-processing (punctuator) and what variation in the number of sentences was observed due to the

**Table 1. Side-by-side in punctuation in an excerpt from Good Girl (Carrie Underwood).**

| DET (12 sentences) | LEM (5 sentences) | WAT (9 sentences) |
|---|---|---|
| But he's really good at lying, yeah, he'll leave you in the dust 'cause when he says forever, well, it don't mean much. Hey, good girl! So good for him. Better back away honey. You don't know where he's been. Why? Why you gotta be so blind? Won't you open up your eyes? It's just a matter of time 'til you find he's no good, girl, no good for you. You better get to getting on your goodbye shoes and go, go, go, yeah, yeah, yeah. He's low. Yeah, yeah, yeah! | But he's really good at lying; yeah, he'll leave you in the dust, 'cause when he says forever, well, it don't mean much. Hey good girl, so good for him, better back away honey, you don't know where he's been. Why, why you gotta be so blind? Won't you open up your eyes, it's just a matter of time, 'til you find he's no good, girl, no good for you, you better get to getting on your goodbye shoes, and go! Go, go, yeah yeah yeah, he's low, yeah yeah yeah. | But, he's really good at lying. Yeah, he'll leave you in the dust 'cause when he says forever, well it don't mean much! Hey good girl so good for him, better back away honey. You don't know where he's been! Why, why you gotta be so blind. Won't you open up your eyes, it's just a matter of time 'til you find he's no good girl. No good for you, you better get to getting on your goodbye shoes and go, go, go. Yeah, yeah, yeah he's low. Yeah, yeah, yeah. |

song style (genre). The null hypotheses being tested are that the parameters $\beta_1$ and $\beta_2$ are equal to 0.

In the model being fitted here, an important underlying assumption is that the mean number of counts between songs is equal to the variation in the number of counts between songs. However, the mean is considerably smaller than the variance for this data. The variance being higher than expected is called overdispersion.

We conducted a test for overdispersion, where the null is that the ratio of the observed variance and the expected variance under the model is 1 (which they would be if they are the same). The test statistic value is 410.5 with a p-value less than 0.001.

The test provides strong evidence that we do indeed have overdispersion in the counts. We address the issue of overdispersion by fitting a quasi-Poisson generalised linear model, which fits an additional dispersion parameter to account for the extra variance. It was estimated to be 4.83, which tells us that the variance is nearly five times greater than the mean.

An Analysis of Variance (ANOVA) table for a two-way factorial model (see Table 2) allows us to assess the importance of each of the two factors separately.

In the first line of Table 2 we can see the sums of squares, degrees of freedom, and F test statistic for the test where we are checking how useful the factor `punctuator` is without considering the other factor. That is the two models being compared are a model with only the Punctator as a predictor and the null model with no predictor variables.

The F test statistic is a standardised measure of the improvement in fit (based on the residual sums of squares) divided by the extra number of parameters needed for the more complex model. The larger the value of the test statistic, the more evidence we have that the more complex model is better.

In the second row, we see the same details for the test of the second factor `genre` when adjusting for the factor `punctuator`. That is, here the two models being compared is the model with only the factor `punctuator` as a predictor variable and the model with both factors.

The p-values in Table 2 show that both factors are useful in terms of predicting the number of sentences. This infers there are significant differences in the number of sentences due to both pre-processing as well as song type. The latter is to be expected considering the three song genres are very different in their sentence composition by design.

We can look at the coefficients, shown in Table 3, to quantify the differences.

Country songs punctuated by DET (the baseline group) are expected to have an average of about exp(3.54) = 34.47 sentences. On average, we can expect to see exp(0.47) = 1.6 and exp(0.97) = 2.64 times as many sentences in pop and rap songs in comparison to country songs, respectively.

More importantly, the expected number of sentences for a song processed by LEM and WAT is exp(-0.56) = 0.57 and exp(-0.43) = 0.65 times the expected number of sentences in a

**Table 2. ANOVA table for a two-way factorial model.**

|  | Sum Sq | Df | F value | P-value |
|---|---|---|---|---|
| Punctuator | 241.75 | 2 | 25.0 | <2.8e-09 |
| Genre | 600.79 | 2 | 62.2 | <2.2e-16 |
| Residuals | 410.50 | 85 |  |  |

As we have fitted a generalised linear model, the output in the ANOVA table is calculated using Type-II Wald tests, where the differences of Wald statistics are used.

**Table 3. (Quasi)Poisson generalized linear model.**

| Coefficients | Estimate | Std Error | t value | P-value |
|---|---|---|---|---|
| Intercept | 3.54 | 0.088 | 40.04 | <2e-16 |
| PunctuatorLEM | -0.56 | 0.086 | -6.51 | 4.93e-09 |
| PunctuatorWAT | -0.43 | 0.082 | -5.16 | 1.60e-06 |
| GenrePop | 0.47 | 0.10 | 4.63 | 1.33e-05 |
| GenreRap | 0.97 | 0.09 | 10.41 | <2e-16 |

song processed by DET, respectively. That is, song scripts processed by DET result in notice-ably more sentences, on average, than both LEM and WAT.

Overall, the most important finding here is that we see significant differences in the number of sentences between the three punctuators, which in turn has implications for cohesion measures that rely on the sentence as a unit.

## 5.2 Correspondence analysis

Correspondence analysis was used to visualize the relationship between the three punctuators and the three song genres in terms of their effect on sentence counts. A Procrustes analysis was used to determine whether differences in the number of sentences between punctuators was due to differences in how they process the data, and not for example, due to intervening variables like exhaustion.

The plot we see in Fig 1 shows a reduced space plot from a correspondence analysis. This analysis is a graphical display of the rows and columns of a contingency table, where the row variable is the punctuator and the column variable the song genre. In the plot, punctuators are labelled DET, LEM, and WAT, while songs are indicated by genre (R = rap, C = country, P = pop) and number as described in section 4.1.

The degree of closeness to each other is of interest, that is, whether two points lie in the same quadrant, as well as their relative distance to the origin. Two points close together would be considered similar in terms of the number of sentences they have.

In Fig 1, we see along Dimension 1, on the horizontal axis, a gradient that captures the level of "verbosity" in the songs that is genre-typical. Rap songs, which tend to have more words, and therefore sentences, are positioned more to the left, while country songs lean towards the right-hand side of the origin. We also see that this highlights the fact that the punctuators WAT and LEM have in common that they tend to punctuate in a manner that leads to fewer sentences. 60.4% of the variation in the number of sentences between songs is explained by the genre.

Along Dimension 2, on the vertical axis, the positioning of the points is driven by the "direction of disagreement" between the two punctuators, in particular LEM and WAT. The songs that differentiate the three punctuators along this second gradient are two rap songs (R8, R9), country song C5 and pop song P9. In these cases, songs where WAT assigned more sentences are above the average and songs where LEM assigned more sentences are below average. 24.6% of the variation in the number of sentences is explained by this contrasting of the punctuators.

Overall, the plot shows the ways in which the three punctuators differ from each other in relation to differences between the songs. That is, while most of the variation in the number of sentences is caused by underlying differences in the type of songs being analysed, we also have nearly a quarter of the variation being caused by differences in how the lyrics were processed.

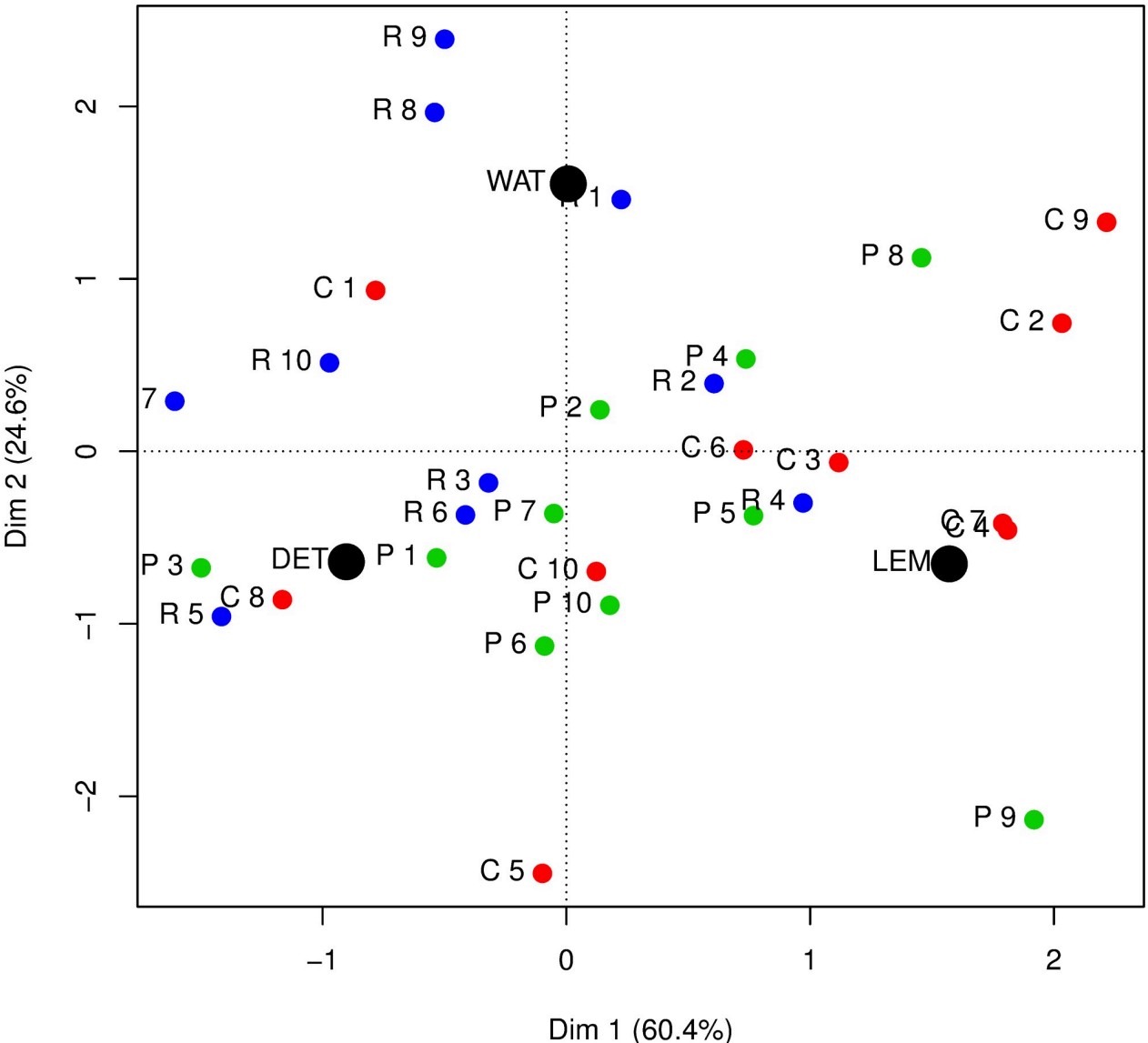

**Fig 1. Reduced space plot from the correspondence analysis.**

## 5.3 Procrustes analysis

Fig 2 shows the results of a Procrustes analysis. A Procrustes analysis plot allows us to focus on the punctuators in the analysis in two respects: First, it shows the degree of similarity between any two persons. In this regard, it is apparent that the three punctuators deviate quite clearly from each other as they are spaced in different quadrants in the plot.

Second, this plot also visualizes the degree of similarity in which processing was performed across the three genres of songs for each individual. To see more detail we need to zoom in, see Fig 3.

The close-up in Fig 3 reveals arrows corresponding to the deviations in the average number of sentences for each type of song from the centroid of the punctuator labelled as LEM. The

## General Procrustes Analysis map

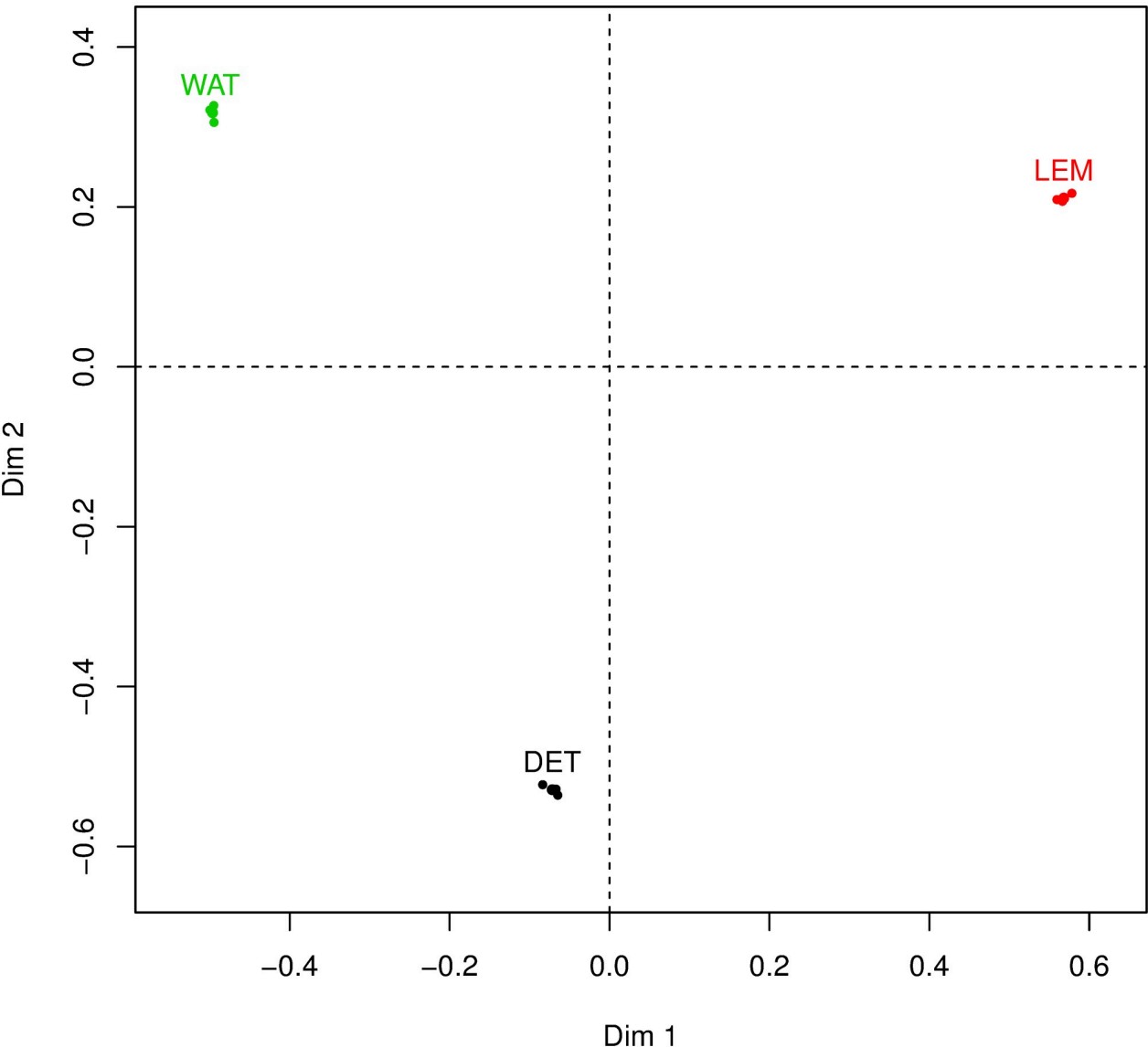

**Fig 2. General procrustes analysis map.**

centroid here is the vector of the mean number of sentences for that punctuator. The fact that these deviations are so small indicates that LEM was consistent in themselves in the way they punctuated across genres.

This observation holds for the other two punctuators as well. This implies that we can assume there is very little noise, caused by, for example, the level of exhaustion the punctuator might experience. That is, the differences in the number of sentences appear to be inherently due to how each of the puncuator processed the lyrics.

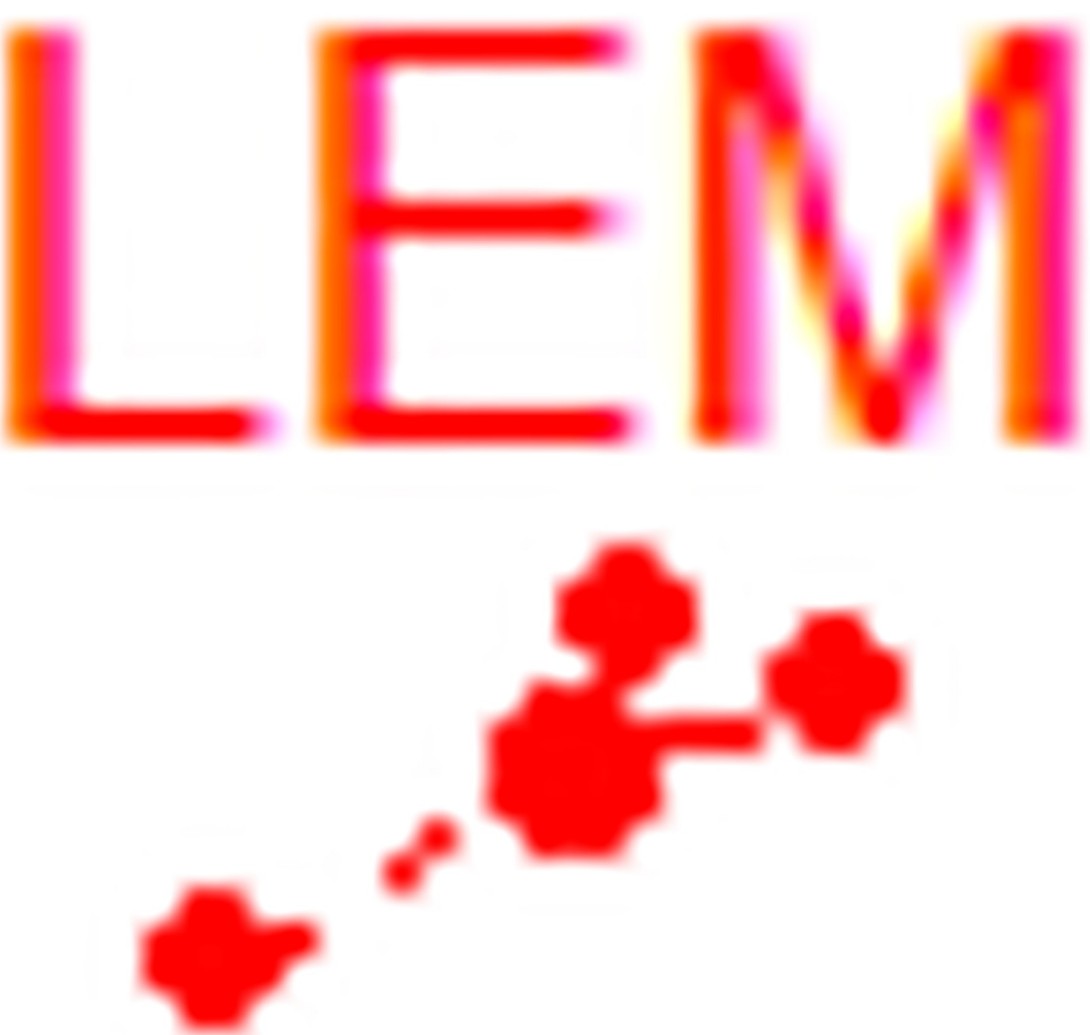

**Fig 3. Close-up of deviations for each genre from LEM's centroid.**

## Discussion

The present study analysed the potential impact of text normalisation on results of the text-analysis program Coh-Metrix 3.0. The focus was on the sentence as a text-structural unit, which bears relevance for a large number of measures provided by the program. In particular, we investigated the impact of punctuation, given Coh-Metrix's apparent reliance on typographic symbols (full stop, exclamation mark, question mark) to detect sentence boundaries. Our first research question asked whether the use of different transcribers to insert punctuation into pop song lyrics can result in significant variance in sentence segmentation. Further, we investigated whether this potential variance in sentence segmentation significantly affects Coh-Metrix measures.

To answer these questions, we quantified the impact of text normalisation at the pre-processing stage on the resulting number of sentences. The impact is found to be significant given the differences found between punctuators in the Poisson regression model (with the total

number of sentences as the response variable) and their visual separation in space in the reduced space plot from the correspondence analysis. We have, thus, demonstrated on a small scale that differences in how a text is pre-processed can lead to a significant difference in the number of sentences. This has implications for at least some Coh-Metrix measures such as sentence length/mean number of words (DESSL) or paragraph length/mean number of sentences (DESPL). As these descriptive measures significantly contribute to a range of other, more sophisticated measures, it can be assumed that these might be affected as well. Consequently, it seems that normalisation of non-traditional written texts that do not clearly indicate sentence boundaries should receive greater attention during planning, analysis and documentation.

The potentially significant influence of transcriber(s) on language data has received greater attention in interview-based qualitative research, where the "nature of transcription as an interpretive activity" has increasingly been acknowledged, along with the fact "that different people checking transcription quality may generate different versions of the interview transcript" ([29], p. 305). [29] highlights the issue of sentence segmentation in originally spoken language and the potential impact on the meaning expressed: "Because people often talk in run-on sentences, judgment calls must be made in the course of transcription about where to begin and end sentences. The insertion of a period or comma can sometimes alter the interpretation of the text" (p.297).

It seems that this argument is equally valid for quantitative research approaches relying on the quantification of sentences rather than or in addition to their meaning in the target data. The need for judgment calls to be made by the transcribers (or punctuators) remains the same. Such judgment calls are not solely based on semantic considerations but are also influenced by grammar and style and each transcriber's attitude towards these matters. As [11] highlight: "The term 'grammar' is indeterminate in the sense that 'What counts as a grammatical English sentence?' is not always a question which permits a decisive answer; (...) questions of grammatical acceptability inevitably become involved with questions of meaning, with questions of good or bad style, with questions of lexical acceptability, with questions of acceptability in context, etc." (p. 47). As the present study has shown, transcribers can answer these questions of meaning, style, and acceptability in different ways, and this can result in significantly different approaches to sentence segmentation. Such differences can be observed despite a similar language background (here: native speakers of New Zealand English) and many years of educational and professional experience with the English language and writing, including at tertiary level. It can be argued that the particularities of a non-standard written genre like song lyrics contribute to these differences. Song lyrics display characteristics of both written and spoken text, sitting "somewhat uneasily on the boundary between writing and speech" ([30], p. 37). Features that might render uniform text interpretation and segmentation difficult include a higher number of disfluencies such as false starts, marginal word, and non-lexical vocables.

While the results of the present study point to the possibility of subjective bias being introduced in automated language evaluation due to punctuation, we note that the sample of 30 songs and only three punctuators is small and presents a limitation. Our work so far represents a preliminary study, and a follow-up study on a larger set of songs and punctuators will be necessary to confirm its findings.

## Conclusion

Coh-Metrix is a sophisticated application that allows both researchers and language practitioners to assess text not only at the word-level but also regarding more global textual features that contribute to the reading ease or difficulty of different texts for various readers, including L2 learners. As Coh-Metrix relies on punctuation for sentence segmentation, caution should be

applied, however, when assessing non-traditional written texts including song lyrics that require the insertion of punctuation during pre-processing. Given the potentially significant impact of a transcriber or punctuator on the language data and consequently on the analysis results, their work should not rely on undefined linguistic intuitions and an assumed consensus regarding punctuation, sentence structure and acceptability, and sentence meaning. Instead, it seems prudent to lay out clear rules and guidelines to be applied during data processing and to make these explicit in sufficient detail in research documentation and publication. Any further research in this area, including replication studies, would benefit from a detailed description of text preparation and normalisation.

## Supporting information

**S1 Dataset. 30 songs were punctuated by each of the three punctuators.** The pre-processed texts were then entered into Coh-Metrix to produce a report which includes the measure "number of sentences". Our data set is a collation of the number of sentences for each song for each punctuator.
(CSV)

**S1 Code. R code used to fit the Poisson regression model, and do the correspondence and Procrustes analyses.**
(R)

## Acknowledgments

We want to express our gratitude to our three punctuators, DET, LEM, and WAT. We would also like to thank the two reviewers for their well thought out and insightful comments.

## Author Contributions

**Conceptualization:** Friederike Tegge, Katharina Parry.

**Data curation:** Friederike Tegge, Katharina Parry.

**Formal analysis:** Katharina Parry.

**Funding acquisition:** Friederike Tegge.

**Visualization:** Katharina Parry.

**Writing – original draft:** Friederike Tegge, Katharina Parry.

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
