## [Decision Letter · Decision Letter 0]

7 Sep 2020

PONE-D-20-05019

The impact of text segmentation on automated evaluation of song-lyrics

PLOS ONE

Dear Dr. Parry,

Thank you for submitting your manuscript to PLOS ONE. After careful consideration, we feel that it has merit but does not fully meet PLOS ONE’s publication criteria as it currently stands. Therefore, we invite you to submit a revised version of the manuscript that addresses the points raised during the review process.

We look forward to receiving your revised manuscript.

Kind regards,

Diego Raphael Amancio

Academic Editor

PLOS ONE

Journal Requirements:

2. We note you have included a table to which you do not refer in the text of your manuscript. Please ensure that you refer to Table 3 in your text; if accepted, production will need this reference to link the reader to the Table.

Reviewers' comments:

Reviewer's Responses to Questions

**Comments to the Author**

1. Is the manuscript technically sound, and do the data support the conclusions?

Reviewer #1: Yes

Reviewer #2: Yes

2. Has the statistical analysis been performed appropriately and rigorously? 

Reviewer #1: Yes

Reviewer #2: Yes

3. Have the authors made all data underlying the findings in their manuscript fully available?

Reviewer #1: Yes

Reviewer #2: Yes

4. Is the manuscript presented in an intelligible fashion and written in standard English?

Reviewer #1: Yes

Reviewer #2: Yes

5. Review Comments to the Author

Reviewer #1: This study addresses an interesting problem that has been overlooked in previous studies on song-lyrics analysis, namely the one of punctuating (in particular of identifying sentence boundaries in) song-lyrics. Previous studies have relied on manual annotations often made by the authors themselves without questioning the objectivity of the task and the fact that different annotators may come up with significantly different annotations.

The present study shows that the task is indeed partly subjective and that annotations do vary from one annotator from another, even between annotators with a strong background or experience in language studies and linguistics. This result is definitely interesting and suggests that researchers should devote more time to punctuating song-lyrics.

The study is well conducted, with a clear and precise analysis. The only slight drawback is the relatively small size of the sample considered (30 songs in total) and the relatively small number of annotators involved (3). If the conclusions drawn are valid, it would have been nice to obtain them on a larger set of songs and annotators.

Reviewer #2: I read the paper “The impact of text segmentation on automated evaluation of song-lyrics” with great interest. In this paper, the authors use three transcribers to punctuate pop, rap and country song lyrics and highlight the resulting differences with respect to some statistics of the number of sentences.

I would like to raise some general concerns:

i) It is not clear for me the role of Coh-Metrix in the paper. Is it used only for counting sentences? There is a whole corpus of introduction mentioning the advantages and the importance of pre-processing before the use of Coh-Metrix, but at line 317 the authors say that their analysis focus only on the number of sentences, instead of “more sophisticated Coh-Metrix measures” (line 318).

ii) In line 329, it is said “Note that we have overdispersion in the counts”. The reader has some trouble to follow this line of thought because we do not have access to the distributions of the number of counts. I would suggest presenting this distribution or, for instance, the inclusion of a figure with the song id in the x-axis (from 1 to 30) and the number of sentences in the y-axis: this figure would have 3 curves, each one indicating the number of sentences obtained from the punctuation of each transcriber. In this example, the dispersion of the data could be seen in the range of y.

iii) The authors should be more descriptive about the tests realized. I believe section 5 could be expanded for the sake of better comprehension. For instance, although ANOVA tables are pretty common, it would be nice to guide the reader to how it is obtained (in a few words) and to the meaning of each column presented in table 2, as well as the relevance of the results to the purpose of the paper.

iv) The authors could explain why they decided to use the Poisson generalized linear model. I mean, is it expected the number of sentences (NoSen) to follow a Poisson distribution?

In conclusion, it is expected that changes in text segmentation would result in differences on text metrics such as the number of sentences. In my opinion, the most interesting aspect of the paper is the quantification of the different interpretations of the same songs: as pointed out in subsection 4.2.1, the transcribers were instructed to follow their “gut feeling” (line 293). In this sense, the results measure how the same set of song can be perceived, felt, seen, viewed, by the transcribers. The importance of this point is already mentioned by the authors when they talk about the ESL (lines 16, 450), but I would suggest an emphasis in the aspect of “quantification of the differences in perception”, including the title and abstract, to make the paper stronger.

6. PLOS authors have the option to publish the peer review history of their article (what does this mean?). If published, this will include your full peer review and any attached files.

Reviewer #1: No

Reviewer #2: No

---

## [Author Response · Author response to Decision Letter 0]

13 Sep 2020

Please see the Response to Reviewers letter. Will provide the latex file here:

%%%%%%%%%%%%%%%%%%%%%%%%%%%%%%%%%%%%%%%%%

\\documentclass[11pt, a4paper]{letter} % Set the font size (10pt, 11pt and 12pt) and paper size (letterpaper, a4paper, etc)

\\input{structure.tex} % Include the file that specifies the document structure

%\\longindentation=0pt % Un-commenting this line will push the closing "Sincerely," and date to the left of the page

%----------------------------------------------------------------------------------------

% YOUR INFORMATION

%----------------------------------------------------------------------------------------

\\Who{Response to Reviewers} % Your name

\\Title{} % Your title, leave blank for no title

\\authordetails{by\\\\

 Dr. Katharina Parry\\\\

 Dr. Friedrike Tegge\\\\

}

%----------------------------------------------------------------------------------------

% HEADER CONTENTS

%----------------------------------------------------------------------------------------

\\logo{logo.png} % Logo filename, your logo should have square dimensions (i.e. roughly the same width and height), if it does not, you will need to adjust spacing within the HEADER STRUCTURE block in structure.tex (read the comments carefully!)

\\headerlineone{} % Top header line, leave blank if you only want the bottom line

\\headerlinetwo{MASSEY UNIVERSITY} % Bottom header line

%----------------------------------------------------------------------------------------

\\begin{document}

%----------------------------------------------------------------------------------------

% TO ADDRESS

%----------------------------------------------------------------------------------------

\\begin{letter}{

}

%----------------------------------------------------------------------------------------

% LETTER CONTENT

%----------------------------------------------------------------------------------------

\\opening{Dear Reviewers,}

as a general comment we want to express our sincere gratitude to you both for taking the time to provide feedback on our work. These are difficult times so we are very appreciative of your efforts, thank-you.

\\textbf{Response to reviewer 1:}

\\textbf{Original comment:} 

\\textit{The study is well conducted, with a clear and precise analysis. The only slight drawback is the relatively small size of the sample considered (30 songs in total) and the relatively small number of annotators involved (3). If the conclusions drawn are valid, it would have been nice to obtain them on a larger set of songs and annotators.}

Thank you for pointing this out. We have added the following two paragraphs:

\\begin{enumerate}

 \\item In the Methods section:

 "We note that the sample of 30 songs and three annotators is small. The work presented here is a first step in a larger project assessing the readability of pop songs for ESL learners. It is intended to address research-methodological problems that seem to be highly relevant not only to the wider investigation at hand but also to the use of Coh-Metrix and automated text evaluation more widely."

 \\item At the end of the Discussion section:

 "While the results of the present study point to the possibility of subjective bias being introduced in automated language evaluation due to punctuation, we note that the sample of 30 songs and only three punctuators is small and presents a limitation. Our work so far represents a preliminary study, and a follow-up study on a larger set of songs and punctuators will be necessary to confirm its findings."

\\end{enumerate}

\\textbf{Response to reviewer 2:}

\\textbf{Original comment:}

\\textit{i) It is not clear for me the role of Coh-Metrix in the paper. Is it used only for counting sentences? There is a whole corpus of introduction mentioning the advantages and the importance of pre-processing before the use of Coh-Metrix, but at line 317 the authors say that their analysis focus only on the number of sentences, instead of “more sophisticated Coh-Metrix measures” (line 318).}

The question as to the relevance of the sentence is a central one. Sentence count is itself a measure, but more importantly, it is an underlying contributor to a large range of more sophisticated measures. As such, we have investigated the differences in sentence count between three different punctuators and draw the conclusion that more sophisticated measures relying on sentence count as a contributing measure would be affected by significant differences in sentence count (such as word overlap between sentences, semantic overlap between sentences, latent semantic analysis, to name a few). This is explained in the section “The relevance of the sentence in Coh-Metrix 3.0”. 

\\newpage

The section includes the following explanation:

Quote from lines 46-68: "It can be speculated that lexical information such as diversity measures, word frequency, word meaningfulness and concreteness, would be largely unaffected by sentence segmentation, whereas certain syntactic indices such as sentence syntax similarity of adjacent sentences (SYNSTRUTa) clearly rely on the sentence as a unit. 

Similarly, cohesion measures such as noun-, argument-, stem-, and content-word overlap in adjacent sentences or across all sentences in a text require sentence boundaries to be indicated, as do measures of Latent Semantic Analysis (LSA), that is, measures of relative semantic overlap between sentences and paragraphs. 

The same holds for traditional readability scores included in Coh-Metrix 3.0, the Flesch Reading Ease (RDFRE) and the Flesch-Kincaid Grade Level (READFKGL), which both rely on sentence length (mean number of words per sentence) to compute text difficulty. 

The Coh-Metrix L2 Readability score (L2), intended to assess the suitability of texts for second language learners, includes content-word overlap in and syntactic similarity of sentences as variables in its formula.

Finally, Coh-Metrix 3.0 offers so-called Text Easability Principal Component Scores, based on a principal components analysis (PCA) of 54 indices in a corpus of written texts found across school-grade levels (K-12) and academic subjects (see Graesser (2011), McNamara(2014) for further details). Five of these principal components – narrativity, syntactic simplicity, word concreteness, referential cohesion, and deep cohesion, account for 54\\% of the variance in the test corpus and are provided to Coh-Metrix users as comprehensive scores that are more easily accessible and of more immediate practical use for teaching practitioners.

Graesser (2011) show that sentence-based measures contribute to three of these easability scores (narrativity, syntactic simplicity, referential cohesion)." 

Furthermore, the argument of the sentence as an underlying measure contributing to a range of other measures is emphasised in our Discussion so that it connects with the following quote:

Quote from lines 406-415: "We have, thus, demonstrated on a small scale that differences in how a text is pre-processed can lead to a significant difference in the number of sentences. This has implications for at least some Coh-Metrix measures such as sentence length/mean number of words (DESSL) or paragraph length/mean number of sentences (DESPL). As these descriptive measures significantly contribute to a range of other, more sophisticated measures, it can be assumed that these might be affected as well. Consequently, it seems that normalisation of non-traditional written texts that do not clearly indicate sentence boundaries should receive greater attention during planning, analysis and documentation."

\\textbf{Original comment:}

\\textit{ii) In line 329, it is said “Note that we have overdispersion in the counts”. The reader has some trouble to follow this line of thought because we do not have access to the distributions of the number of counts. I would suggest presenting this distribution or, for instance, the inclusion of a figure with the song id in the x-axis (from 1 to 30) and the number of sentences in the y-axis: this figure would have 3 curves, each one indicating the number of sentences obtained from the punctuation of each transcriber. In this example, the dispersion of the data could be seen in the range of y.}

Well spotted, this was certainly something we needed to explain better in order for the reader to not just take our word for it, but be able to see it for themselves. Thank you for the helpful suggestion for a plot which would show the overdispersion. However, a plot takes up a lot of space, and we weren't sure if the reader would know what to look for (as in, we would need to give some explanation as well) so we instead decided to do test for overdispersion as another way of getting the reader to understand how we detect overdispersion. 

That is, in section 5, as highlighted, we explain the idea of the test, e.g. that the test statistic is the ratio of the expected variance (under the model) and the observed variance. We then use the highly significant p-value as proof that we should consider a quasi-Poisson model instead to account for the difference in the true variance from the variance assumed under the model.

\\textbf{Original comment:}

\\textit{iii) The authors should be more descriptive about the tests realized. I believe section 5 could be expanded for the sake of better comprehension. For instance, although ANOVA tables are pretty common, it would be nice to guide the reader to how it is obtained (in a few words) and to the meaning of each column presented in table 2, as well as the relevance of the results to the purpose of the paper.}

Once again, following this advice will improve the article greatly, and as such we have added clarification in section 5 as highlighted. 

\\textbf{Original comment:}

\\textit{iv) The authors could explain why they decided to use the Poisson generalized linear model. I mean, is it expected the number of sentences (NoSen) to follow a Poisson distribution?}

This is correct, we do indeed treat the number of sentences as counts, and thus assume they are Poisson distributed. Clearly, this was not explained well enough in the current form of the article and thus the explanation has now been given more attention when addressing the previous comment.

\\textbf{Original comment:}

\\textit{In conclusion, it is expected that changes in text segmentation would result in differences on text metrics such as the number of sentences. In my opinion, the most interesting aspect of the paper is the quantification of the different interpretations of the same songs: as pointed out in subsection 4.2.1, the transcribers were instructed to follow their “gut feeling” (line 293). In this sense, the results measure how the same set of song can be perceived, felt, seen, viewed, by the transcribers. The importance of this point is already mentioned by the authors when they talk about the ESL (lines 16, 450), but I would suggest an emphasis in the aspect of “quantification of the differences in perception”, including the title and abstract, to make the paper stronger.

}

Our research shows that different transcribers delineate the structural unit of the sentence differently. However, our research does not provide insight into whether these decisions are made based on semantic interpretation (felt, seen, viewed) or solely on different views on the definition of the sentence unit, on sentence grammar, acceptability and style. It can be assumed that it is a combination of these factors, but that assumption cannot be inferred from the present results. 

What our study shows is that there can be differences in text segmentation between educated transcribers and these differences can have an impact on the results of automated quantification of textual features. Based on a review of quantitative studies involving Coh-Metrix and also literature describing the role of the sentence in NLP, we can also state that this problem does not seem to have received any attention. 

Based on your thoughtful feedback we have updated the title to "The impact \\textbf{of differences in} text segmentation on \\textbf{the} automated \\textbf{quantitative} evaluation of song-lyrics" as well as the abstract as highlighted in the revised manuscript.

\\bigskip

Again, thanks so much for your input!

Sincerely, \\par \\medskip

Friederike and Katharina 

\\end{letter}

\\end{document}

---

## [Decision Letter · Decision Letter 1]

26 Oct 2020

The impact of differences in text segmentation on the automated quantitative evaluation of song-lyrics

PONE-D-20-05019R1

Dear Dr. Parry,

We’re pleased to inform you that your manuscript has been judged scientifically suitable for publication and will be formally accepted for publication once it meets all outstanding technical requirements.

Kind regards,

Diego Raphael Amancio

Academic Editor

PLOS ONE

Additional Editor Comments (optional):

Reviewers' comments:

Reviewer's Responses to Questions

Comments to the Author

1. If the authors have adequately addressed your comments raised in a previous round of review and you feel that this manuscript is now acceptable for publication, you may indicate that here to bypass the “Comments to the Author” section, enter your conflict of interest statement in the “Confidential to Editor” section, and submit your "Accept" recommendation.

Reviewer #2: All comments have been addressed

2. Is the manuscript technically sound, and do the data support the conclusions?

Reviewer #2: Yes

3. Has the statistical analysis been performed appropriately and rigorously? 

Reviewer #2: Yes

4. Have the authors made all data underlying the findings in their manuscript fully available?

Reviewer #2: (No Response)

5. Is the manuscript presented in an intelligible fashion and written in standard English?

Reviewer #2: Yes

6. Review Comments to the Author

Reviewer #2: (No Response)

7. PLOS authors have the option to publish the peer review history of their article (what does this mean?). If published, this will include your full peer review and any attached files.

Do you want your identity to be public for this peer review?

 For information about this choice, including consent withdrawal, please see our Privacy Policy.

Reviewer #2: No

---

## [Editor Report · Acceptance letter]

28 Oct 2020

PONE-D-20-05019R1 

The impact of differences in text segmentation on the automated quantitative evaluation of song-lyrics 

Dear Dr. Parry:

I'm pleased to inform you that your manuscript has been deemed suitable for publication in PLOS ONE. Congratulations! Your manuscript is now with our production department. 

Kind regards, 

on behalf of

Dr. Diego Raphael Amancio 

Academic Editor

PLOS ONE